# The Macro- and Micro-Mechanics of the Colon and Rectum I: Experimental Evidence

**DOI:** 10.3390/bioengineering7040130

**Published:** 2020-10-19

**Authors:** Saeed Siri, Yunmei Zhao, Franz Maier, David M. Pierce, Bin Feng

**Affiliations:** 1Department of Biomedical Engineering, University of Connecticut, Storrs, CT 06269, USA; siri@uconn.edu (S.S.); yunmei.zhao@uconn.edu (Y.Z.); david.pierce@uconn.edu (D.M.P.); 2Lightweight Design and Composite Materials, University of Applied Sciences Upper Austria, 4600 Wels, Austria; franz.maier@uconn.edu; 3Department of Mechanical Engineering, University of Connecticut, Storrs, CT 06269, USA; 4Department of Physiology and Neurobiology, University of Connecticut, Storrs, CT 06269, USA

**Keywords:** colorectum, large intestine, mechanotransduction, multi-layered, experiments

## Abstract

Many lower gastrointestinal diseases are associated with altered mechanical movement and deformation of the large intestine, i.e., the colon and rectum. The leading reason for patients’ visits to gastrointestinal clinics is visceral pain, which is reliably evoked by mechanical distension rather than non-mechanical stimuli such as inflammation or heating. The macroscopic biomechanics of the large intestine were characterized by mechanical tests and the microscopic by imaging the load-bearing constituents, i.e., intestinal collagen and muscle fibers. Regions with high mechanical stresses in the large intestine (submucosa and muscularis propria) coincide with locations of submucosal and myenteric neural plexuses, indicating a functional interaction between intestinal structural biomechanics and enteric neurons. In this review, we systematically summarized experimental evidence on the macro- and micro-scale biomechanics of the colon and rectum in both health and disease. We reviewed the heterogeneous mechanical properties of the colon and rectum and surveyed the imaging methods applied to characterize collagen fibers in the intestinal wall. We also discussed the presence of extrinsic and intrinsic neural tissues within different layers of the colon and rectum. This review provides a foundation for further advancements in intestinal biomechanics by synergistically studying the interplay between tissue biomechanics and enteric neurons.

## 1. Introduction

The large intestine refers to the segment of the gastrointestinal (GI) tract distal from the Ileocecal valve to the anal verge, consisting of ascending, transverse, descending, sigmoidal colon and the rectum. The physiological functions of the large intestine include absorbing water, moving waste residue down the GI tract, and temporary fecal storage, all of which involve mechanical movement and deformation of the tubular GI structure. Pathophysiological conditions in the large intestine are often associated with the malfunction of intestinal biomechanics, including changes in gut motility and muscle activities (e.g., irritable bowel syndrome [IBS] [1], chronic constipation [2], Hirschsprung’s Disease [3], intestinal neuronal dysplasia [4]), changes in tissue structure and compliance (e.g., inflammatory bowel disease [5], diverticula diseases [6], intestinal tuberculosis [7]), luminal shear flow (e.g., chronic idiopathic diarrhea [8]), and voluntary muscle controls (e.g., fecal incontinence [9]). In addition, chronic visceral pain from the colon and rectum has a prominent mechanical component—it is mechanical distension, not heating, pinching, cutting, or inflammation that reliably evokes pain from hollow visceral organs [10]. Motivated by providing a further mechanistic understanding of those disease conditions, the biomechanics of the large intestine, especially the colon and rectum has been under investigation since the 1970s (e.g., [11,12,13,14,15]).

The wall of the large intestine can be divided into four anatomically distinct layers (from inner to outer): mucosa, submucosa, muscularis propria (inner circular and outer longitudinal layers), and serosa. Nonetheless, most studies did not consider this through-thickness heterogeneity when characterizing the macroscopic biomechanics of the colon and rectum. An array of conventional mechanical testing methods have been applied to characterize the macroscopic mechanical features of the large intestine, including uniaxial tensile stretch [16,17,18], biaxial tensile stretch [19,20,21], planar compression [22,23], indentation [24,25], shear [22], and luminal inflation [26,27,28,29]. Overall, the mechanical response of the large intestine is governed by the geometrical (large strains in vivo) and material nonlinearities of the tissue with its embedded bundles of collagen and muscle fibers. The tangential stiffness, i.e., the slope of the stress–strain curve increases almost exponentially with increasing strain in the physiologically relevant loading range, indicating gradual recruitment of embedded fibers with progressive deformation. Mechanically, the large intestine is anisotropic (e.g., stiffer in the longitudinal direction than in circumferential direction [20]), viscoelastic [30], and contractile [31]. Only two studies conducted mechanical tests on the large intestine, considering it as a layered biological structure [20,32], and these indicate that the mechanical strength of the tissue is mainly determined by the submucosa and muscular layers whereas serosa and mucosa have no significant stiffness.

The microscale mechanics of the large intestine can correlate to the content and morphology of collagen and muscle fibers within the wall tissue, which provides structural support to almost all soft biological tissues [33]. Intestinal collagen fibers were classically determined by chromatic or antibody-based staining, which was usually limited to thin tissue sections (e.g., [34,35,36,37]). Recent advances in multiphoton microscopy allowed imaging collagen fibers by second harmonic generation, i.e., a nonlinear, label-free imaging modality that allows precise and selective detection of collagen fibers deep into soft biological tissues [38]. The microstructures of the collagen fibers in sublayers of the intestinal wall collectively inform the macroscopic mechanical behaviors in bulk intestinal tissues. In addition, the network of collagen fibers determines the local mechanical environment, which could profoundly impact the neuronal tissues embedded in the intestinal wall (e.g., [39]). Both intrinsic and extrinsic neurons in the intestine play critical roles in the aforementioned mechanical events, including gut motility [40], muscle activities [41], and mechanotransduction [10]. It is worth mentioning that nerve endings are concentrated in the submucosa and muscular layers [40,42], which are the main load-bearing structures of the intestinal wall [20,32]. Hence, determining the micromechanics of the large intestine, especially its interplay with intestinal neuronal tissues will likely advance our understanding of the biomechanical factors in physiological and pathophysiological conditions in the large intestine.

Many GI disorders in the large intestine are poorly managed in the clinics. Especially, chronic visceral pain is the cardinal complaint of patients with IBS that affects up to 20% of the U.S [43]. population. Despite its prevalence, managing IBS-related visceral pain remains an unmet clinical need. The intestinal biomechanics plays critical roles in driving noxious perception from the large bowel. Hence, to further advance the mechanistic understanding of various GI disorders, especially visceral pain, we have conducted this timely and comprehensive review to summarize the experimental evidence on macroscopic and microscopic biomechanics of the large intestine.

## 2. The Layered Structure and Function of Large Intestine

The large intestine extends approximately 150 to 180 cm in adults and consists of mostly the colon and rectum. Compared to the small intestine, the colon is larger in diameter, mainly fixed in position, and plays a very limited role in nutrient absorption. The main functions of the colon are to push the waste content down the GI tract by coordinated wave-like mechanical movement (peristalsis), absorb water to form solid feces, and send them to the rectum, which is the distal six inches of the GI tract. The large intestine is also host to a significant amount of bacteria, archaea and fungi, i.e., the intestinal microbiota, which perform a host of useful functions such as fermenting unused energy substrates, training the immune system, and producing vitamins and hormones, etc. The rectum’s main functions are continence and defecation, i.e., temporarily storing feces and expelling feces out of the body, respectively. As shown in Figure 1, the large intestine comprises four main layers: mucosa, submucosa, muscularis propria (longitudinal and circumferential muscle layers), and serosa. Each layer has distinct anatomical structures and neuronal tissue contents and consequently serves different biomechanical role in intestinal physiology and pathophysiology. We briefly review this layered anatomic structure of the large intestine as follows.

The mucosa is the inner lining of the colon and rectum, consisting of a thin layer of epithelium, a connective tissue layer (i.e., lamina propria), and a thin layer of muscle (muscularis mucosa). Lack of a prominent role for nutrient absorption by the large intestine has resulted in the absence of the villi structure in the mucosa, i.e., small, folded components that greatly enhance the intraluminal surface area of the small intestine. The epithelium in the large intestine “sinks” into large cylindrical structures, i.e., the crypts (Figure 1) that extend through the lamina propria to the muscularis mucosa. Due to the enhanced surface area by the crypts, the epithelium in the large intestine is the largest epithelial barrier of the body [45], where the secretory and absorptive processes (mostly water) take place [46]. Sensory nerve endings extend to the laminal propria between the crypts and act as “taste buds” of the gut to survey the mucosal contents [47].

The submucosa is a fibrous connective tissue layer that surrounds the mucosa. It contains major branches of blood and lymph vessels supplying the large intestine. It has a high concentration of lymphocytes, fibroblasts, and mast cells. In addition, the submucosa hosts one of the two major enteric neural plexuses, i.e., the submucosa plexus that regulates the configuration of the luminal surface, controls glandular secretions, alters electrolyte and water transport, and regulates local blood flow. We recently reported the concentrated presence of thick collagen fibers within the submucosa via second harmonic generation imaging [44], which provides anatomic support for its load-bearing function, as determined from layer-separated tensile tests [20,32]. In addition, amongst all four layers of the colon, submucosa has the highest proportion (32%) of extrinsic sensory neural endings, according to a recent tracing study [42]. Moreover, we recently showed that most sensory endings in the submucosa are free endings, small in diameter (~1 micron), and meandering, which are anatomic features of typical nociceptors that encode tissue-injurious mechanical stimuli [44].

The muscularis propria consists of two layers of smooth muscle: an inner circular muscle layer and an outer longitudinal muscle layer. Between these two muscular layers is the myenteric plexus, one of the two major enteric neural plexuses. The ganglia in the myenteric plexus are more prominent than their submucosal counterpart. The myenteric plexus receives projections from preganglionic parasympathetic fibers as well as postganglionic sympathetic fibers [40]. The myenteric plexus directs the coordinated wave-like propulsive movement of the colon (peristalsis) via a “hardwired” polarized polysynaptic peristaltic circuit [44]. Despite its relatively thin through-thickness dimension (~2% of the total wall thickness of mouse colon from the authors’ unpublished observations), the myenteric plexus also hosts 22% of the extrinsic sensory nerve endings [42]. The circular muscle layer has a slightly greater amount of sensory nerve endings (25%) but is significantly thicker [42]. In contrast, there are barely any extrinsic sensory innervations in the longitudinal muscle layers [42].

The serosa is the outermost layer of the intestinal wall and is composed of a continuous sheet of squamous epithelia cells, the mesothelium [44]. It represents an extension of the visceral peritoneum and mesentery as it envelops the intestine. Although the serosa has a significant content of collagen fibers [44], it is unlikely to have a major load-bearing role due to its thin thickness compared to the total intestinal wall thickness [32]. Like the longitudinal muscle layer, the serosa also lacks significant sensory innervation [42,44].

## 3. Macroscale Biomechanics of the Colon and Rectum

The macroscale biomechanics of the large intestine was characterized mostly by uniaxial and biaxial extension and luminal inflation, and to a lesser extent by compression, shear, and indentation tests. Biaxial tensile stretch and luminal inflation best match the mechanical conditions in vivo. These conventional means were extensively implemented for studying the esophagus and small intestine. Despite the prevalence of lower-GI disorders and the importance of large intestine biomechanics in those disorders, the body of literature devoted to studying the macroscale biomechanics of the large intestine is smaller than those for the upper GI tract. In Table 1, we summarize studies that reported outcomes of mechanical tests on colonic and rectal tissues. Most studies were conducted in vivo on human distal colon and rectum via inflation tests. For in vitro tests conducted on excised tissues, uniaxial tensile tests were most widely used. The vast majority of the studies considered the intestinal wall as a homogeneous “thick” membrane and applied mechanical tests on complete tissue from the bulk wall. In contrast, only two studies considered the layered wall structure and conducted mechanical tests on separated layers [20,32]. The large intestine shares similar biomechanical features as other soft biological materials like blood vessels and skin, all of which can be considered as an incompressible, hyperelastic composite of fibers embedded in a viscoelastic base material. Considering the large intestine as a cylindrical tube, we assign the cylindrical coordinates as shown in Figure 2.

In the longitudinal direction, the large intestine changes its geometry slightly, gradually decreases in diameter and increases in thickness from the proximal to distal regions. Radially, the large intestine is heterogeneous showing four major layers (see Figure 1). In addition, the large intestine is vascularized through the mesentery that is aligned longitudinally with one of the three taenia coli, indicating structural heterogeneity along the circumferential direction. The biomechanical heterogeneities along those three directions are summarized and discussed below.

### 3.1. Biomechanical Heterogeneity along the Longitudinal Direction, from Colon to Rectum

Most mechanical tests were conducted on the distal portion of the large intestine, especially the sigmoidal colon and rectum. The passive mechanical properties, i.e., in the absence of muscle activities, were systematically characterized at four different locations along the rat large intestine by two independent studies, both of which reported the highest mechanical stiffness in the transverse colon [26,60]. From the transverse colon, circumferential stiffness decreases progressively towards both the distal and proximal region of the large intestine, with the proximal colon showing the lowest circumferential stiffness of all. The longitudinal stiffness is comparable between the proximal and transverse colon and decreases progressively towards the distal colon and rectum. We recently conducted biaxial tensile tests and reported higher compliance in the rectum than in the distal colon of mice [19], consistent with data reported above on rat tissues. However, one study reported that the biomechanical properties are comparable in human sigmoid colon and rectum in vivo [50]. This likely reflects the presence of smooth muscle tone resulting in increased rectal stiffness under physiological conditions.

In mouse distal colon and rectum, we systematically studied the longitudinally heterogeneous behaviors in pre-stress, tissue stiffness, viscoelasticity, and anatomic thickness [20]. We reported a gradual increase in opening angles from the distal colon towards the rectum, indicating increased residual strain in rectal regions in physiological conditions which could potentially contribute to higher firing rates of sensory never endings in the rectum versus the colonic counterparts [61]. From the opening angle in layer-separated studies, we reported compressive residual stress in the mucosa/submucosa composite, and tensile residual stress in the serosa and muscular layers. The distal colon and rectum are predominately innervated by sensory afferents from the lumbar splanchnic nerve (LSN) and the pelvic nerve (PN), respectively. Interestingly, these differential biomechanical properties between the distal colon and rectum correspond with differential sensory neural encoding in those regions. The distal colon is less compliant in the circumferential direction compared to the rectum, which is consistent with the predominant presence of sensory nerve endings in the PN innervation that encode circumferential intestinal stretch. This is also consistent with lower firing rates of colonic sensory endings to circumferential stretch than rectal endings [61]. The colorectal tissue is viscoelastic and dissipates more energy under deformation in the circumferential direction than in the longitudinal, which could explain the adaptation of afferent activities to circumferential intestinal stretch [61,62]. Finally, the rectum is significantly thicker than the colon [61], especially at the circular muscular layers, which implies that PN sensory endings are more affected by smooth muscle activities during normal GI functions than the LSN counterparts.

### 3.2. Biomechanical Heterogeneity along the Radial Direction

The layered structure of the intestinal wall strongly indicates heterogeneous biomechanical properties through the thickness of the wall, i.e., along the radial direction. Most biomechanical studies considered the colon as a homogeneous membrane and carried out the tests on the whole colorectal wall. Egorov et al. [32] conducted studies on layer-separated large intestinal tissues harvested from the human cadavers and reported that the mechanical strength of the bowel wall is determined by the submucosa and muscular layers while the serosa and mucosa have no significant strength. However, no technical details were provided regarding how different intestinal layers were separated and tested. We recently reported a layer-separated biomechanical study on mouse distal colon and rectum. In mouse large intestine, there is an apparent interstitial space between the submucosa and circular muscle layers. This unique anatomic feature allowed us to conduct fine dissections to gently separate the intestinal wall into inner and outer composites, the inner consisting of the mucosa and submucosa, and the outer of the two muscular layers and serosa [20]. We reveal that the inner mucosal/submucosal composite has slightly higher longitudinal stiffness than the outer composite, while the outer muscular/serosal composite has higher circumferential stiffness. Hence, the wall tension resulting from colorectal distension will be undertaken by both composites with the inner composite taking slightly more longitudinal tension and the outer composite more circumferential tension. Therefore, studies by others and us strongly indicate that the mechanical stiffness of the bowel wall is determined by the submucosa and muscular layers whereas the serosa and mucosa have no significant stiffness.

### 3.3. Biomechanical Heterogeneity along the Circumferential Direction

Unlike the tubular organs such as the blood vessels that are usually axisymmetric along the central axis, the presence of mesentery along one side of the colorectum makes it a nonhomogeneous tissue additionally in the circumferential direction. Moreover, the distribution of LSN sensory endings in the circumferential direction is not homogeneous and is only concentrated close to the mesentery [62]. In human and porcine colons, the longitudinal muscle layers are not continuously distributed along the circumference but concentrated in three bands, i.e., taeniae coli. One of the taeniae coli connects with the mesentery and is termed the mesenteric taeniae, whereas the other two are termed anti-mesenteric taeniae [63]. All this anatomic evidence strongly suggests the differential biomechanical properties along the circumferential direction between the mesenteric and non-mesenteric zones. In colons from patients with diverticular disease, small sacs or pockets are predominantly developed in regions next to the mesenteric taeniae [63,64], which provides indirect evidence to suggest the weaker mechanical strength at mesenteric regions versus non-mesenteric regions in the large intestine. In support, our preliminary unpublished observations indicate that mouse colorectum ruptures at the mesenteric zone when intraluminal pressure is beyond 200 mmHg.

### 3.4. In-Plane Biomechanical Anisotropy

The colorectum can be considered as a thin-walled cylinder in which the longitudinal and circumferential directions form a planar surface. As reported by several studies [19,22,26,32], the in-plane mechanical properties of the colorectum in those principal directions are quantitatively different. All studies confirm that the colorectum has in-plane tissue anisotropy, indicating higher stiffness in the longitudinal direction than in the circumferential direction. Higher circumferential compliance facilitates the physiological function of the large intestine of fecal storage and propagation, while lower longitudinal compliance reduces longitudinal deformation to keep the large intestine in position during distension [26]. In addition, this tissue anisotropy between the longitudinal and circumferential directions is more pronounced in either the proximal or distal end of the large intestine. Both the circumferential and longitudinal stiffness is the highest in the middle segment of the large intestine, i.e., the transverse colon [26,60]. Circumferential stiffness is significantly lower both at the proximal colon and rectum. In the meanwhile, longitudinal stiffness is only modestly reduced at the proximal colon and rectum. This collectively results in significant in-plane anisotropy in the proximal colon and rectum. The increased circumferential compliance likely supports its physiological role of fecal storage. Moreover, enhanced circumferential compliance in the proximal colon may facilitate the coupling of the large intestine with the small intestine, and particularly with its most distensible segment, the ileum [26].

### 3.5. Macroscopic Mechanical Tests on Large Intestines with Lower GI Disorders

The macroscopic mechanical characterization was also conducted on colon and rectum from patients with lower GI disorders or in animal disease models. Increased colon stiffness was reported in most inflammatory bowel disease conditions [5] with a few exceptions when the inflammation is ongoing and more localized in the mucosa (e.g., active ulcerative colitis [48,51]). Colons from patients with diverticular diseases showed reduced mechanical strength, reduced distensibility, and premature relaxation to distension [11,48], along with prominent structural changes of thickened circular muscular layer, shortened colon length, and narrowed lumen [4]. Hirschsprungs’ disease features loss of ganglionic neurons in the colon and rectum, increased circular muscle thickness, and increased intraluminal pressure during colonic inflation [3]. In contrast, no apparent change in colonic or rectal biomechanics was reported in patients with irritable bowel syndrome [28]. In summary, change in macroscopic biomechanics of the large intestine is evident in most of the lower GI disorders except for functional disorders like the irritable bowel syndrome.

## 4. Microscale Experimental Evidence (Large and Small Intestine)

The microscale networked collagen fibers form the major load-bearing skeleton for many biological tissues, e.g., skin [65], tendon and cartilage [66], and blood vessels [67], and also determines the macroscale mechanical properties of the colorectum. Collagen fibers have diameters ranging from 0.5 to a few microns and they are assembled from thread-like collagen fibrils [68]. Collagen fiber morphology in the small and large intestine was determined by a handful of studies on sectioned tissue slices using chromatic and immunological staining and scanning electron microscopy [69,70,71,72,73,74]. Sokolis and Sassani used light microscopy to inspect the orientation of muscle, elastin, and collagen fibers [27], indicating that the configuration of collagen network differs greatly across the sublayers of large intestinal wall. Second-harmonic generation (SHG) microscopy has emerged as a powerful method for imaging collagen fibers with submicron resolution in a diverse range of tissues, which is highly selective to the collagen fibril/fiber structure and can visualize collagen fibers several hundred microns deep into the tissue using excitation light at infra-red range (800–1200 nm). SHG imaging on large intestinal tissues was reported by several recent studies in the literature [75,76,77,78,79,80,81,82,83,84,85]. Compared with conventional staining methods on thin tissue slides of ~10 microns thick, we recently reported that SHG microscopy can visualize collagen fibers through the thickness of an intact mouse colon (~200 microns thick) and most of the rectum (300–400 microns) [46]. SHG allows systematic characterization of the collagen fiber density, distribution, alignment, and orientation at different layers of the colorectum [86]. Most literature on micromechanical studies of large intestine was limited to the superficial mucosal layer. In contrast, the micromechanics of the deeper layers in large intestine were rarely reported. We have thus included micromechanical studies from both large and small intestine in this review and summarized in Table 2.

As mentioned previously, the intestine from lumen to the outside is composed of four main layers: mucosa, submucosa, muscularis propria, and serosa. Other than our recent study [44], there are no reports in the literature that systematically characterized the collagen fiber contents and orientations through the wall thickness of either small or large intestine. We conducted SHG imaging to quantify the thickness and the diameter of fibers of each distinct through-thickness layer of the colorectum, as well as the principal orientations, corresponding dispersions in orientations, and the distributions of diameters of collagen fibers within each of these layers. We found that collagen fibers are concentrated in the submucosal layer. The average diameter of collagen fibers was greatest in the submucosal layer, followed by in the serosal and muscular layers. Collagen fibers aligned with muscle fibers in the two muscular layers, whereas their orientation varied greatly with location in the serosal layer.

Mucosa: Most studies on intestinal collagen have focused on the pathology of the mucosa to detect or differentiate various intestinal diseases such as cancer [77,80,81,84,93], dysplasia [76,81], colorectal lesions [83], and inflammatory bowel diseases (IBD) [79]. The collagen fibers in the mucosal layers appear to orient circularly by wrapping around individual colonic crypts [27,86], which are invaginated tube-like structures in the mucosa. Collagen fibers there do not seem to form an in-plane network like their counterparts in the submucosa, and thus are unlikely to play a significant load-bearing role in resisting colorectal distension. However, mucosal collagen fibers likely provide structural support of the crypt structures which are innervated by afferent endings [42]. The biomechanical role of collagen fibers in the mucosa is unclear and awaits further experimental studies. It may contribute to colorectal mechanotransduction by translating mucosal shearing into local mechanical stress/strain around the crypts.

Submucosa: A few studies have focused on this intestinal layer to assess its morphology, fibers angle, and collagen content in healthy [27,70,73,74,75,80] and diseased [71] subjects. Collagen fibers concentrate in the submucosa of the colorectum, as determined by SHG imaging. More significantly, collagen fibers in the submucosa are wavy when the colorectum is in load-free condition and can gradually straighten with increasing distension of the colorectum [44]. This recruitment of collagen fibers agrees with the tension–stretch relations recorded from the mucosal/submucosal composite which shows increased mechanical stiffness with deformation [20]. Under noxious distension, the collagen fibers in the submucosa straighten to reveal two principal groups of fibers that orient approximately plus and minus 30° from the longitudinal direction, respectively (similar to the observation in [19,69,70,86]). The two groups of fibers appear to not run in parallel planes but interweave with one another to form a reinforced network of collagen fibers. In addition, the collagen fiber network in the submucosa does not seem to vary significantly in thickness, fiber density, or fiber diameter from proximal to distal colorectum [20,86]. This likely contributes to the consistent longitudinal stiffness in tension despite the significant increase in colorectal wall thickness from colonic to rectal regions. Consistent observations between the microscopic collagen fiber network and bulk mechanical properties from biaxial tensile tests strongly indicate that the submucosa is the load-bearing ‘skeleton’ for the colorectum and protects it from excessive distension.

Muscularis propria: The collagen fibers in the two muscular layers are generally characterized by chromatic or immuno-staining studies on thin tissue sections. Our SHG study through the thickness of the colon and rectum allows direct comparison of relevant collagen contents in the muscle layers as compared to other intestinal layers [86]. Compared with submucosa and serosa, the circular muscular and longitudinal muscle layers are low in collagen fiber contents. However, the outer muscular/serosal composite shows comparable longitudinal and circumferential stiffness as the inner mucosal/submucosal composite, indicating the contribution of muscle fibers to the mechanical stiffness in the outer composite. The orientations of collagen fibers in the two muscular layers are well aligned with the muscle fiber directions, i.e., longitudinal and circumferential, respectively. Those two groups of collagen fibers perpendicular to one another collectively lead to the reduced in-plane tissue anisotropy in the outer muscular/serosal composite as compared with more pronounced tissue anisotropy in the inner mucosal/submucosal composite [20].

Serosa: the serosal layer as a connective tissue membrane is also rich in collagen [86], but its contribution to the macroscale mechanical strength of the colorectum is limited by its thinness. The thickness of the serosa is usually no more than 50 μm in human colon [32].

### Altered Collagen Morphology and Contents in Lower GI Disorders

The majority of the studies in Table 2 focus on characterizing changes in intestinal collagen under disease conditions in the lower GI, especially in the mucosal layer. Pathophysiological structure and content of collagen fibers in large intestine were mainly studied in the following lower GI disordered: Crohn’s disease (CD), Ulcerative colitis (UC), intestinal tuberculosis (ITB), and colonic cancer/dysplasia. Compared with healthy controls, UC patients have more frequent occurrence of defects in the collagen stroma and fibroblasts of the colonic mucosa, significantly decreased mucosal blood flow, and more severe multiple platelet agglutinations within the small mucosal vessels [94]. Furthermore, collagen contents in the mucosa were reported to differentiate between CD and ITB; mucosal collagen content in ITB is significantly higher than in CD [82]. Moreover, CD affects the collagen distribution in clusters while ITB affects collagen around the perimeter of caseating granulomata. In normal colon and rectum, there is a dense matrix of collagen fibers almost parallel to the interface of epithelium and stroma, which in patients with colonic dysplasia showed a loosened matrix with a tilted angle to the interface [76,81]. Therefore, the collagen density and the collagen fiber direction underneath the intestinal epithelium could be used to discriminate between normal and dysplastic colonic mucosa. In addition, malignant tumor is reported to alter the organization of collagen fibers in the colonic mucosa even 10 cm to 20 cm away from its location. This tumor causes an increase in collagen fiber thickness which correlates with the formation of invadopodia, i.e., specialized cell structures that increase the migratory capacity of cancer cells [85]. In contrast, changes in collagen contents in intestinal layers other than the mucosa are rarely reported.

## 5. Conclusions and Outlook

The macro- and micro-scale biomechanics of the colon and rectum have significant implications in lower GI disorders. Various mechanical behaviors of the colon and rectum are altered in disease conditions, including gut motility and muscle activities, tissue structure and compliance, luminal shear flow, and voluntary muscle controls. The macroscale biomechanics of the colon and rectum were systematically characterized by conventional test methods for soft biological tissues, revealing the large intestine’s hyperelastic and viscoelastic material properties similar to other fiber-reinforced biological tissues. The large intestine varies in diameter and thickness from proximal to distal regions, consists of four major layers in the wall, and is circumferentially unsymmetrical due to the mesenteric attachment along one side of the intestine. These anatomical features also correlate with heterogeneous biomechanical properties along the longitudinal, radial, and circumferential directions of the tubular large intestine. The large intestine is stiffest in the middle segment, i.e., the transverse colon, and more compliant at either the proximal or the distal ends, i.e., the proximal colon and rectum. Increased rectal compliance facilitates fecal storage, while increased compliance in the proximal colon allows better mechanical coupling with the ilium, the most distensible segment of the small intestine. Overall, the large intestine is stiffer longitudinally than circumferentially, reducing elongation during physiological distension and peristalsis to allow the intestine to maintain position. Through the wall thickness, the submucosa and the muscularis propria are the major load-bearing structures whereas the mucosa and serosa have no significant stiffness. Along the circumferential direction, regions close to the mesentery are mechanically softer than regions away from the mesentery as suggested by the concentrated distribution of sacs or pockets in the mesenteric region of the colon in patients with diverticular diseases.

The microscale biomechanics of the large intestine is reflected by the contents, morphology, and orientation of collagen fibers in different layers of the large intestine. Collagen is concentrated in the submucosa and serosa whereas the content in the muscularis propria and mucosa is significantly lower. In addition, collagen fibers in the submucosa have the largest average diameter and consist of two groups of fibers oriented approximately ±30° along the longitudinal direction to form a tight-knit network. This microstructure of collagen fibers in the submucosa indicates its load-bearing roles to support substantial mechanical loads. Despite low contents of collagen fibers, muscularis propria showed a mechanical stiffness comparable to the submucosal/mucosal composite, and its load-bearing role is likely provided by the thick muscle bundles in the circular and longitudinal muscle layers. Extrinsic and intrinsic neural innervations concentrate in the submucosa and muscularis propria, which are focal regions of high mechanical stresses during physiological intestinal distension and peristalsis. The nociceptor-like nerve endings in the submucosa strongly indicate their critical roles in detecting tissue-injurious mechanical stimuli by evoking pain from the colon and rectum. In addition, changes in collagen content and morphology in intestinal mucosa can potentially be a marker for many lower GI disorders.

Recent years have seen significant progress in our biomechanical understanding of the large intestine in both health and diseases, especially from recent applications of the SHG imaging to characterizing the collagen fibers in intestinal tissues. Further advancement in the field of intestinal biomechanics will likely take place in close association with knowledge of neural innervations of the large intestine, which plays critical roles in an array of mechanical events in the large intestine like peristalsis, smooth muscle tone, and mechano-nociception. The fact that the load-bearing region of the large intestine (submucosa and muscularis propria) correlates with the concentrated distribution of intestinal neuronal tissues is likely not a coincidence. Understanding the synergistic interplay between the intestinal biomechanics and neurophysiology will advance our mechanistic understandings of various lower GI disorders in which biomechanical factors play critical roles. In addition, accelerated application of SHG detection of collagen contents and morphology in the intestinal mucosa will likely reveal changes in the microscale mechanics in various lower GI disorders. Future studies will likely extend the detection beyond the mucosal layer to reveal changes in collagen of deeper layers of the large intestine. Last, the integration of multimodal experimental evidence by multiscale computational simulation of both the intestinal tissue biomechanics and the encoding of neuronal tissues in the intestinal wall will likely drive theoretical hypotheses about various lower GI disorders to help elucidate their etiology.

## Figures and Tables

**Figure 1 bioengineering-07-00130-f001:**
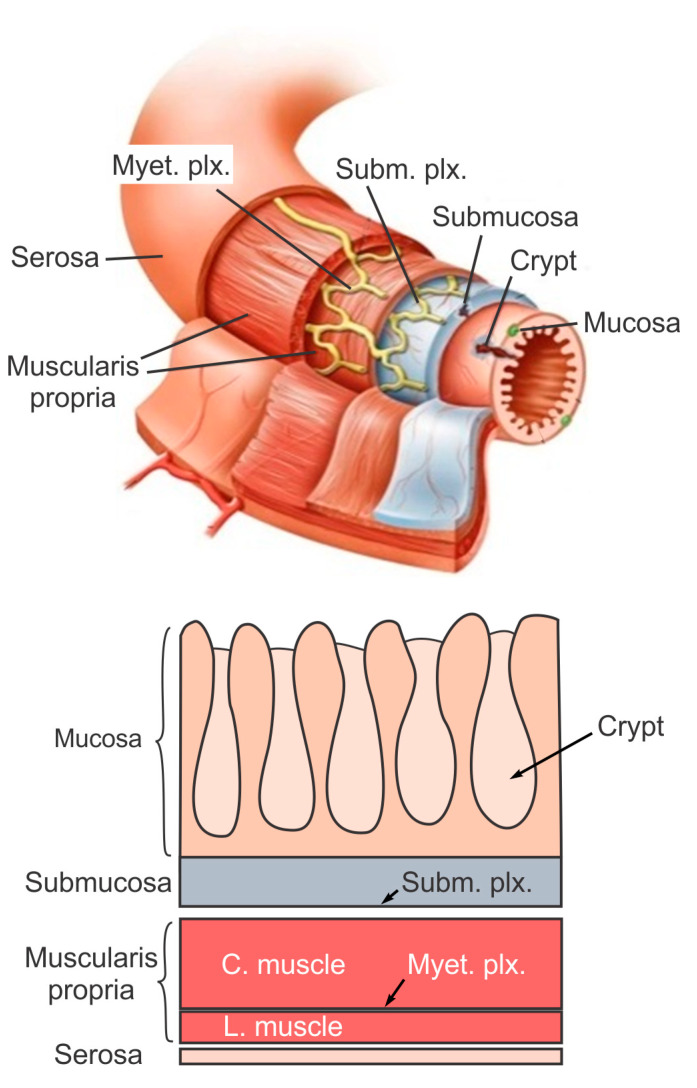
The layered structure of the large intestine. Subm. plx. = submucosal plexus; Myet. plx. = myenteric plexus; C. muscle = circular muscle layer; L. muscle = longitudinal muscle layer. Adapted from [44] with permission.

**Figure 2 bioengineering-07-00130-f002:**
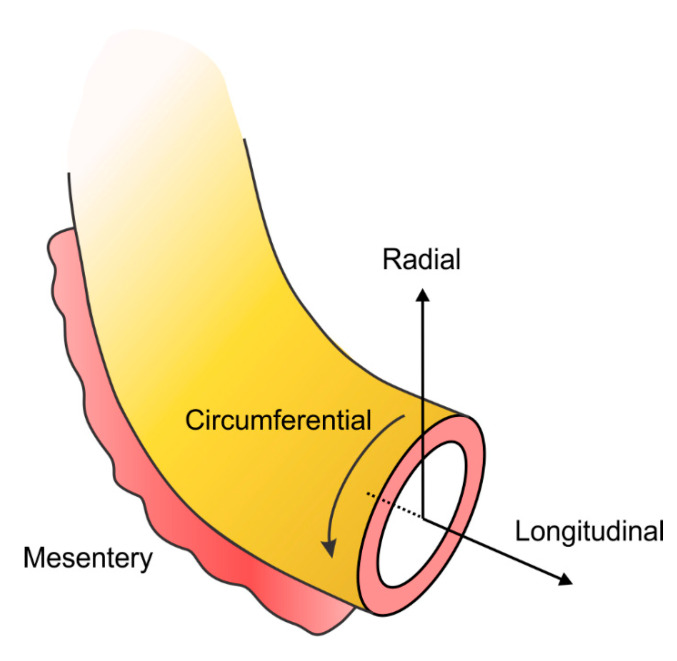
Schematic diagram showing the cylindrical coordinates of the large intestine in circumferential, radial, and longitudinal directions.

**Table 1 bioengineering-07-00130-t001:** Mechanical tests on large intestinal tissues.

Specimen	Condition	Test Methods	References
***Mechanical tests on whole intestinal wall***			
Human	In vivo	Pressure-volume	Parks, 1970 [11]; Smith et al., 1981 [48]; Bharucha et al., 2001 [30]
		Pressure-CSA (Cross-section area)	Arhan et al., 1976 [13]; Dall et al., 1993 [49]; Drewes et al., 2001 [28]; Petersen et al., 2001 [50]; Drewes et al., 2006 [51]
Human	In vitro	Uniaxial stretch	Watters et al., 1985b [15]; Glavind et al., 1993 [52]; Massalou et al., 2016 [53]; Massalou et al., 2019 [54]; Massalou et al., 2019 [55]
		Biaxial stretch	Howes and Hardy, 2012 [56]
Porcine	In vitro	Inflation	Carniel et al., 2015 [57]; Patel et al., 2018 [29,57]
		Compression and shear	Qiao et al., 2005 [22]
		Uniaxial stretch	Qiao et al., 2005 [22]; Ciarletta et al., 2009 [58]; Carniel et al., 2014 [16]; Christensen et al., 2015 [59]
		Biaxial extension	Puértolas et al., 2020 [21]
Goats	In vitro	Compression	Higa et al., 2007 [23]
Rats	In vitro	Pressure-diameter	Gao and Gregersen, 2000 [60]; Sokolis et al., 2011 [26]; Sokolis and Sassani, 2013 [27]
		Uniaxial stretch	Watters et al., 1985 [14]
		Indentation	Stewart et al., 2016 [24]
Mice	In vitro	Biaxial stretch	Siri et al., 2019 [19]
***Mechanical tests on separated intestinal layers***			
Mice	In vitro	Biaxial stretch	Siri et al., 2019 [20]
Human	In vitro	Uniaxial stretch	Egorov et al., 2002 [32]

**Table 2 bioengineering-07-00130-t002:** Quantification of collagen fibers in the small and large intestine.

Specimens	Tissue		Layers		References
***Chromatic and fluorescent staining***					
Human	Large intestine	Mucosa		Zonios et al., 1996 [87]	
Rats	Large intestine	Mucosa		Sokolis and Sassani, 2013 [27]	
		Submucosa		Sokolis and Sassani, 2013 [27]	
		Muscular layers		Sokolis and Sassani, 2013 [27]	
Porcine	Small intestine	Submucosa		Abraham et al., 2000 [88]	
***Small-angle light scattering***					
Porcine	Small intestine	Submucosa		Sacks and Gloeckner, 1999 [89]	
***Polarized light microscopy***					
Rats	Small intestine	Submucosa		Orberg et al., 1983 [70]; Zeng et al., 2003 [73]; Yu et al., 2004 [74]	
***Electron microscopy***					
Human	Large intestine	Mucosa		Donnellan et al., 1966 [90]; Shamsuddin et al., 1982 [45]	
		Submucosa		Thomson et al., 1987 [91]	
Rats	Small intestine	Submucosa		Orberg et al., 1982 [69]; Orberg et al., 1983 [70]; Gabella, 1983 [71]	
Porcine	Small intestine	Submucosa		Gabella, 1983 [71]	
***Two-photon excited fluorescence and second harmonic generation microscopy***					
Human	Large intestine	Mucosa		Zhuo et al., 2011 [76]; Zhuo et al., 2012 [77]; Liu et al., 2013 [78]; Schürmann et al., 2013 [79]; Bianchi et al., 2014 [80]; Birk et al., 2014 [81]; Mao et al., 2016 [82]; He et al., 2019 [83]; Sarri et al., 2019 [84]; Despotović et al., 2020 [85]	
		Submucosa		Jiang et al., 2011 [75]; Bianchi et al., 2014 [80]	
Mice	Large intestine	Mucosa		Xu et al., 2013 [92]; Prieto et al., 2019 [93]; Maier et al., 2020 [44]	
		Submucosa		Maier et al., 2020 [44]	
		Muscular layers		Maier et al., 2020 [44]	
		Serosa		Maier et al., 2020 [44]	

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
