# Peer review of "The Macro- and Micro-Mechanics of the Colon and Rectum I: Experimental Evidence"

_bioengineering, 2020, doi:10.3390/bioengineering7040130_

Round 1
Reviewer 1 Report
The submitted manuscript is a well written and easily understandable review of experimental evidence of the macro- and micromechanics of the colon and rectum.
Sections 1 and 2 provide a thorough (to the extent that this is possible on a complex anatomical subject on 4 pages) overview of the function and anatomy of the large intestine (and rectum). The macroscale biomechanics in Section 3 are likewise presented in a comprehensive manner with easy to grasp explanations and extensive references on prior published mechanical tests in Table 1, which are then systematically presented in the subsequent subsections.
Section 4 on microscale experimental evidence also mixes in information on the small intestine, which I am uncertain is in the scope of a review of the colon and rectum. Of course, from a histological and embryogenesis point of view, the intestine as a whole shares many similarities, however, it is unclear to what extent knowledge gained from the small intestine can be generalized for computational models for the large intestine. I would suggest delineating and expounding further on the extent to which these sources can be assumed to cross-apply to the large intestine.
Finally, the Conclusion section concludes with an intelligible explanation of the importance of a multi-scale model incorporating the synergistic interplay between biomechanics and neurophysiology.
Over all, with the caveat that the inclusion of small intestine experimental information needs to be better justified as it does not strictly fit the scope of the article, a well done review article that can be accepted with minor revisions.
Author Response
Response to reviewer 1
The submitted manuscript is a well written and easily understandable review of experimental evidence of the macro- and micromechanics of the colon and rectum.
Sections 1 and 2 provide a thorough (to the extent that this is possible on a complex anatomical subject on 4 pages) overview of the function and anatomy of the large intestine (and rectum). The macroscale biomechanics in Section 3 are likewise presented in a comprehensive manner with easy to grasp explanations and extensive references on prior published mechanical tests in Table 1, which are then systematically presented in the subsequent subsections.
We appreciate the reviewer’s commendation on the manuscript!
Section 4 on microscale experimental evidence also mixes in information on the small intestine, which I am uncertain is in the scope of a review of the colon and rectum. Of course, from a histological and embryogenesis point of view, the intestine as a whole shares many similarities, however, it is unclear to what extent knowledge gained from the small intestine can be generalized for computational models for the large intestine. I would suggest delineating and expounding further on the extent to which these sources can be assumed to cross-apply to the large intestine.
We thank the reviewer for the comments. We found limited publications on the micromechanical studies that focused exclusively on the large intestine, especially studies that imaged beyond the superficial mucosal layer of the large intestine. For example, the recent publication from our group was the only one that showed the ±30 degree orientation of collage fibers in the submucosal layer of the large intestine [1]. This is supported by a handful of publications conducted on small intestine [2, 3]. Hence, we felt obligated to include those studies in this review. Due to the limited information on micromechanics of large intestine beyond the mucosal layer, we have included studies on small intestine to provide a more comprehensive profile on intestinal micromechanics in this review.
Finally, the Conclusion section concludes with an intelligible explanation of the importance of a multi-scale model incorporating the synergistic interplay between biomechanics and neurophysiology.
Over all, with the caveat that the inclusion of small intestine experimental information needs to be better justified as it does not strictly fit the scope of the article, a well done review article that can be accepted with minor revisions.
We thank the reviewer for the positive comments on this manuscript and have provided justification to include small intestinal biomechanics above.
References
- Maier, F., et al., The heterogeneous morphology of networked collagen in distal colon and rectum of mice quantified via nonlinear microscopy. Journal of the Mechanical Behavior of Biomedical Materials, 2021. 113.
- Orberg, J., E. Baer, and A. Hiltner, Organization of collagen fibers in the intestine. Connective tissue research, 1983. 11(4): p. 285-297.
- Orberg, J., L. Klein, and A. Hiltner, Scanning electron microscopy of collagen fibers in intestine. Connective tissue research, 1982. 9(3): p. 187-193.
- Sokolis, D.P. and S.G. Sassani, Microstructure-based constitutive modeling for the large intestine validated by histological observations. J Mech Behav Biomed Mater, 2013. 21: p. 149-66.

Reviewer 2 Report
The review article by Siri and colleagues goes to great lengths in describing available experimental observations regarding the macro- and micro-mechanical properties of the large intestine in healthy and pathologic states. Emphasis is placed to the discussion of mechanical heterogeneities of the large intestinal wall (longitudinal, radial=layer-specific, and circumferential) and of the existing imaging methods used to characterize the load-bearing intestinal constituents, namely collagen fibers. The distribution of neurons within the different intestinal layers is also highlighted.
Overall, the topic is interesting and this reviewer is not aware of a previous review article in the field of large intestinal biomechanics, so that the objective of this article is worthwhile. The authors should be congratulated on a well-written manuscript. I have no major suggestions for improvement. Few minor points are provided below:
- Is the upper panel of Fig. 1 drawn by the authors or taken by a book? In the latter case, it must be stated wherefrom the illustration is taken.
- Page 3, lines 107-109: There is repetition of the text "that extend through the lamina propria to the muscularis mucosa". Please, revise.
- Page 3, line 111: "secretary"->"secretory".
- Page 4, line 145: "The macroscale biomechanics of the large intestine were" -> "was".
- Page 9, lines 295-297: Ref. [27] appears to have reported on the layer-specific collagen-fiber contents and orientations along the rat large intestine. Please, discuss.
- Page 12, line 413. "intestinal mucosa reveal changes"->"will likely reveal changes".
Author Response
Response to reviewer 2
The review article by Siri and colleagues goes to great lengths in describing available experimental observations regarding the macro- and micro-mechanical properties of the large intestine in healthy and pathologic states. Emphasis is placed to the discussion of mechanical heterogeneities of the large intestinal wall (longitudinal, radial=layer-specific, and circumferential) and of the existing imaging methods used to characterize the load-bearing intestinal constituents, namely collagen fibers. The distribution of neurons within the different intestinal layers is also highlighted.
Overall, the topic is interesting and this reviewer is not aware of a previous review article in the field of large intestinal biomechanics, so that the objective of this article is worthwhile. The authors should be congratulated on a well-written manuscript. I have no major suggestions for improvement. Few minor points are provided below:
- Is the upper panel of Fig. 1 drawn by the authors or taken by a book? In the latter case, it must be stated wherefrom the illustration is taken.
We thank the reviewer for pointing this out. We have added the source of Fig 1A in the revision [1].
- Page 3, lines 107-109: There is repetition of the text "that extend through the lamina propria to the muscularis mucosa". Please, revise.
We have deleted the redundant sentence as suggested.
- Page 3, line 111: "secretary"->"secretory".
We have corrected the typo as suggested.
- Page 4, line 145: "The macroscale biomechanics of the large intestine were" -> "was".
We have corrected as suggested.
- Page 9, lines 295-297: Ref. [27] appears to have reported on the layer-specific collagen-fiber contents and orientations along the rat large intestine. Please, discuss.
We have added the follow sentence to discuss the paper in page 8.
Sokolis and Sassani used light microscopy to inspect the orientation of muscle, elastin, and collagen fibers [4], indicating that the configuration of collagen network differs greatly across the sublayers of large intestinal wall.
- Page 12, line 413. "intestinal mucosa reveal changes"->"will likely reveal changes".
We have corrected as suggested.
References
- Maier, F., et al., The heterogeneous morphology of networked collagen in distal colon and rectum of mice quantified via nonlinear microscopy. Journal of the Mechanical Behavior of Biomedical Materials, 2021. 113.
- Orberg, J., E. Baer, and A. Hiltner, Organization of collagen fibers in the intestine. Connective tissue research, 1983. 11(4): p. 285-297.
- Orberg, J., L. Klein, and A. Hiltner, Scanning electron microscopy of collagen fibers in intestine. Connective tissue research, 1982. 9(3): p. 187-193.
- Sokolis, D.P. and S.G. Sassani, Microstructure-based constitutive modeling for the large intestine validated by histological observations. J Mech Behav Biomed Mater, 2013. 21: p. 149-66.

Reviewer 3 Report
I found this manuscript interesting as a comprehensive summary of the knowledge regarding biomechanics of the colon.
My biggest concern is related to the claim of being a systematic review. A systematic scientific review should use systematic methods to collect secondary data that are then appraised and synthesized. This has not been done. At least the methodology for finding the literature has not been described and it is obvious that, according to the reference list, there is a strong bias to self-referrals.
Other comments
Abstract:
Lines 20-21. Again, this is not a systematic review. The intro should include that it looks at animal and human models. Writing “health and disease” does not make sense; another suggestion would be “in healthy and diseased tissues”.
Intro:
In the description of the colon, properties related to vitamin absorption and important interactions with microbial population are not mentioned.
Line 81. Please add a reference to the statement of 20% IBS in the US.
Line 88. Please use metric system according to international standards.
Section 4.1:
It seems to me that the authors are describing changes in the submucosa and not the mucosa. Please correct or clarify.
In this manuscript, the information related to diseased tissues is sparse.
Adding information related to changes in inflammatory diseases (other than Chron’s) and infectious diseases (other than tbc) would increase the value of its content.
Line 395: What is the meaning of the word "families", what are their names? Please rephrase or clarify.
Author Response
Response to reviewer 3
I found this manuscript interesting as a comprehensive summary of the knowledge regarding biomechanics of the colon.
My biggest concern is related to the claim of being a systematic review. A systematic scientific review should use systematic methods to collect secondary data that are then appraised and synthesized. This has not been done. At least the methodology for finding the literature has not been described and it is obvious that, according to the reference list, there is a strong bias to self-referrals.
Other comments
Abstract:
Lines 20-21. Again, this is not a systematic review. The intro should include that it looks at animal and human models. Writing “health and disease” does not make sense; another suggestion would be “in healthy and diseased tissues”.
We thank the reviewer for the comments. We have strived to include all the studies on large intestine in this review and believe that the breadth of literature included in this review is sufficient to justify it as a systematic review.
Intro:
In the description of the colon, properties related to vitamin absorption and important interactions with microbial population are not mentioned.
We have included additional sentences to discuss the bacterial environment and vitamin production in the colon (Pages 2 and 3).
Line 81. Please add a reference to the statement of 20% IBS in the US.
We have added the reference 43 for the statement.
Line 88. Please use metric system according to international standards.
We have changed to the SI units.
Section 4.1:
It seems to me that the authors are describing changes in the submucosa and not the mucosa. Please correct or clarify.
In this section, we are describing the changes in the mucosa as it is clearly mentioned in Page 11, Lines 371, 372, and 375.
In this manuscript, the information related to diseased tissues is sparse. Adding information related to changes in inflammatory diseases (other than Chron’s) and infectious diseases (other than tbc) would increase the value of its content.
Following the reviewer’s suggestion, we have added additional sentences in section 4.1 to describe changes in collagen contents under GI disorders.
Line 395: What is the meaning of the word "families", what are their names? Please rephrase or clarify.
We thank the reviewer’s suggestion and have replace the term family with group throughout.
